# Quantitative Evaluation of Ecological Stress Caused by Land Use Transitions Considering the Location of Incremental Construction Lands: The Case of Southern Jiangsu in Yangtze River Delta Region

**Pingxing Li** [1,2], **Chonggang Liu** [1,2] **and Hui Cao** [1,2,*]

1. Nanjing Institute of Geography and Limnology, Chinese Academy of Sciences, Nanjing 210008, China; pxli@niglas.ac.cn (P.L.); cgliu@niglas.ac.cn (C.L.)
2. Key Laboratory of Watershed Geographic Sciences, Chinese Academy of Sciences, Nanjing 210008, China
* Correspondence: hcao@niglas.ac.cn; Tel.: +86-025-86882134

**Abstract:** With their significance in connecting socio-economic development and related eco-environmental consequences, land use transitions have gradually become the focus of land change science and sustainability science. Although various research studies have determined the ecological effects of land use transitions and provided suggestions to regulate them, few studies have investigated the different ecological stress of construction lands from the perspective of their spatial locations in ecologically differentiated regions. Taking economically developed and highly urbanized southern Jiangsu in Eastern China as an example, we developed a process-based method to indicate the spatial heterogeneity of ecological suitability and divided southern Jiangsu into five-level ecological zones accordingly. Considering that construction lands in ecological zones with higher ecological suitability levels cause greater ecological stress, we evaluated the ecological stress levels of incremental construction lands at different stages after 1990. Then, we carried out the calculation of county-level ecological stress and county-level zoning based on both the area and ecological stress level of their incremental construction lands. Results indicated that ecological zones with the highest to lowest ecological suitability levels accounted for 49.85%, 25.73%, 15.56%, 6.51%, and 2.34%, respectively. The majority of the incremental construction lands had the highest and moderately high ecological stress levels, and they were mainly distributed in areas along the Yangtze River and around Taihu Lake. The general ecological stress level of southern Jiangsu was at a relatively high level at each stage, but the county-level patterns of ecological stress levels were spatially different. As determined from the relationship between the amount of incremental construction lands and the average stress level associated with these lands in each unit, four types of zones, i.e., H-H, H-L, L-H and L-L zones, were identified, and targeted suggestions on land use regulations were proposed. We conclude that the spatial distribution of incremental construction lands significantly affects their ecological consequences from the perspective of maintaining ecosystem integrity. Both construction lands and ecological suitability are location specific, so the location-oriented evaluations could provide an effective approach for determining the spatial patterns of land use transitions based on spatially differentiated ecological consequences. It is essential to propose location-specific policies to carry out spatially precise ecological restoration and the redistribution of incremental construction lands.

**Keywords:** land use transitions; ecological stress; ecological process; construction lands; location; southern Jiangsu; China

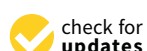



## 1. Introduction

Since the late 1970s, unprecedented urbanization and industrialization have taken place in China [1,2]. According to the China Statistical Year Book in 1987 and 2016, the scale of construction land in China was reported to be 386,000 square kilometers in 2015,

accounting for 4.1% of the total land area, and this area has expanded by 190% since 1986. The rapid expansion of construction lands has not only resulted in various issues, such as social conflicts and economic polarization [3–5] but has also imposed serious challenges associated with food deficits and ecological security [6–8]. Moreover, land use transitions have taken place recently and have become the focus of land use research, especially in rapid developed and urbanized China [9–11]. The ecological effects of recent land use transitions are more diverse and complicated than traditional land use changes [12,13]. Aiming to promote sustainable development and coordinate land development and ecological protection, the current state of research suggests that assessing the spatial stress associated with land use transitions and the expansion of construction lands is urgent for providing insight into policymaking and for regulating land use transitions [14,15].

Existing published research has focused on ecological stress assessments of the expansion of construction lands from various perspectives and at different levels. In early studies, many different indicators and metrics were designed to analyze the state of or spatial and temporal changes in regional ecosystems, including the areas and types of construction lands, ecosystem services, ecological vulnerability and various landscape-level metrics [16–18]. These methods are still widely used with the help of multi-source land use, land cover or ecological/environmental data. The abovementioned studies effectively indicated the ecological consequences of land use transitions from certain perspectives, but the cause of the above ecological consequences i.e., land use transitions, have not been given sufficient attention [19]. For land use transitions, it is widely accepted that the land use morphology includes dominant morphology and recessive morphology [20,21]. The spatial pattern of land use types, especially the location of construction lands, is among the most significant topics of land use transition research [9,19]. However, some direct indicators reflecting the state or change process of land use have been proposed recently, such as the amount and proportion of construction lands [19,22]. Some complicated metrics have been proposed by adopting the amount or proportion of land use as an output-oriented or input-oriented parameter [23,24]. However, the location of construction lands was rarely considered [25,26].

Ecosystems are characterized by integrative and mutually relevant components, so the protection of a specific ecological process is critical for improving the efficiency of ecological protection [27–29]. However, the ecological suitability is spatially differentiated, and different locations in certain ecosystem are of unlikely significance in maintaining ecosystem health [30,31]. Moreover, according to the landscape ecology theory, maintaining ecological processes by creating continuous ecological corridors is of equal or even higher importance than restoring some landscape patches [32,33]. Landscape structures, patterns, and processes are highly interrelated and interactive, and protected ecological processes promote landscape connectivity and avoid fragmentation among isolated patches [34,35] Therefore, the spatial differentiation of ecological suitability should be considered in evaluating the ecological consequences of construction lands [19].

On the other hand, the land suitability for development is influential for the spatial expansion of construction lands, and lands with high development suitability are usually the optimal area for land development and the allocation of construction lands [36,37]. Construction lands might be location at zones with high development suitability before the 21st century, while the ecological suitability of those zones is neglected sometimes, causing obvious ecological stress accordingly [38,39]. The location of construction lands should be emphasized, as the chosen locations are critically significant in maintaining regional ecological safety by safeguarding and controlling ecological processes [25,40]. Although some researchers have carried out studies to indicate the effect of location on the ecological stress associated with construction lands, in these studies, only the effects of construction lands at certain stages were analyzed [19,26]. The expansion of construction lands reflects the spatiotemporal patterns of land use transitions, and targeted studies are still needed to better indicate the different ecological stress levels caused by the spatially differentiated expansion of construction lands.

The ecological stress effect caused by the locations of construction lands is influenced by the spatial heterogeneity of regional ecosystems, and it is widely accepted that construction lands in regions with higher ecological suitability levels cause higher ecological stress to regional ecological safety [26,35]. Accordingly, it is of primary importance to evaluate the spatial heterogeneity of ecological suitability [41,42]. Early studies were mostly conducted with the help of single-element or multifactor assessment approaches, and the importance of ecological processes was not significantly emphasized [25,43]. To reveal the role of these ecological processes, conservation biologists have developed various methods, including biotelemetry, tagged release capture, mass tagged capture, and species surveys, to track the movement of specific species and determine the dispersal processes that promote species migration [44,45]. However, the methods listed above are often difficult and costly at the regional scale, and high-efficiency model simulations with low data requirements have been widely adopted, such as cellular automata (*CA*) models, system dynamics (*SD*) models, and other spatially explicit models, (see, for example, [46–49]). Among these models, the minimum cumulative resistance (*MCR*) model is one of the most widely used simulation models. This model has its roots in the recognition of dispersal processes or "stepping stones" and was recently widely used in ecological process simulations and ecological network constructions [50–52]. Accordingly, spatial differences in ecological suitability have been effectively identified from the perspective of maintaining necessary ecological processes [19,43,53].

Overall, the ecological stress caused by construction lands differs spatially, and these differences are caused by the spatial locations of construction lands and the location-related differentiation of ecological suitability. The spatial conflict between ecological protection and expansion of construction lands existed and brought challenges to regional sustainable development. However, the spatial differentiation of ecological suitability and related ecological stress of construction lands during different stages of land use transitions have not been analyzed. To better solve the above-mentioned issues and promote regional sustainable development by optimizing the spatial pattern of land development, it is urgent to consider ecological process analyses to indicate ecosystem heterogeneity and to consider the location effect of construction lands to assess the associated ecological stress level [54,55].

We chose economically developed and rapidly urbanized southern Jiangsu in Eastern China as an example and tried to answer the following questions: (1) How do we indicate the spatial differentiation of ecological suitability, especially from the perspective of maintaining necessary ecological process? (2) What are the spatial patterns of the expansion of construction lands and how can we evaluate their ecological stress caused by different locations during transitional stages? (3) What are the spatial patterns and temporal changes of differential ecological stress of construction lands? In the following parts, we firstly provide a brief introduction of southern Jiangsu and the adopted methodology, examine the spatial ecological suitability pattern based on the significance of maintaining ecological processes, and then evaluate the differentiated ecological stress levels caused by the presence of incremental construction lands at different stages. Finally, we discuss our potential contributions and policy implications from the perspectives of promoting ecological processes and optimizing the spatial distribution of construction lands.

## 2. Methodology

### 2.1. Study Area

Southern Jiangsu is located in the Yangtze River Delta region (YRD) of Eastern China covers a land area of 28,000 km$^2$. There are five prefectural cities in southern Jiangsu (i.e., Nanjing, Wuxi, Changzhou, Suzhou, and Zhenjiang), and these cities are further divided into 33 county-level units (Figure 1a). It is one of the most developed regions in China [2]. In 2015, it had 33.24 million residents and a regional gross domestic product (GDP) of RMB 4151.87 billion, which were about 1.58 and 14.33 times those in 1990. As population and industries have grown rapidly in this region, construction lands have ex-

panded dramatically (Figure 1b). After 1990, the expansion of construction lands was about 4818 km$^2$, and the total area of construction lands reached 6617 km$^2$ by 2015, accounting for 23.45% of the total regional land area [26]. The urbanization process of southern Jiangsu has experienced several transitional stages influenced by the interactions of industrialization, urbanization, globalization and marketization [40,56]. Land use transitions occurred accordingly, which are represented by the changing spatial and temporal expanding patterns of the construction lands at different stages [57,58]. However, the ecological effects and their differences of construction lands that occur during several different transitional stages are rarely researched, especially from the perspective of differentiated spatial location of construction lands [19]. As the rapid expansion of construction lands has occupied a large amount of ecologically suitable lands and brought obvious ecological destruction and environmental degradation, it is necessary to conduct detailed research and provide insight into the spatial and temporal ecological stress patterns caused by the expansion of construction lands.

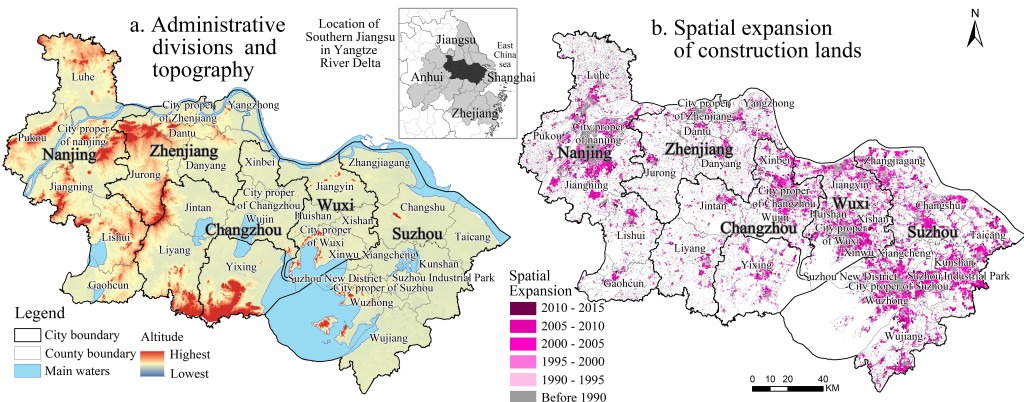

**Figure 1.** Administrative divisions, topography and expansion of construction lands in southern Jiangsu.

### 2.2. Methodologies and Data

#### 2.2.1. Model Description for Ecological Process Simulations

As discussed above, the ecological effects of construction lands are location specific, and more ecologically suitable regions experience higher ecological stress levels when land development occurs there [19]. Moreover, the spatial differentiation of ecological suitability is affected by the importance of maintaining ecological processes. Besides the process of species dispersal, the expansion of ecological spaces is also a typical ecological process, and its simulation is feasible with some originally species-specific methods, including the *MCR* model [48,53]. As the *MCR* values of the above model indicate the cumulative resistance to the expansion of ecological spaces, zones with lower *MCR* values are more suitable as potential ecological spaces [51,59]. Therefore, we first identified ecological suitability through ecological process simulations with the *MCR* model. According to the studies of Knaapen et al. [60], Yu [61] and Li et al. [59], the *MCR* model can be expressed as follows:

$$MCR = f_{\min} \sum\nolimits_{j=n}^{i=m} D_{ij} \times R_i \qquad (1)$$

where *f* is a monotonically increasing function that indicates the least resistant relation between unit *i* and source unit *j* under the restriction of a certain resistance surface, min denotes the minimum cumulative resistance value produced in different processes from unit *i* to unit *j*, $D_{ij}$ is the spatial distance between *i* and *j*, and $R_i$ represents the resistance of cell *i* on the route from unit *i* to *j*. The *MCR* value reflects the minimum cumulative resistance and maximum migration accessibility or ecological spatial expansion ability of species from the source to the target.

Based on the *MCR* analysis, the most accessible paths (i.e., potential ecological processes) between ecological sources and target patches can be identified. The *MCR* values of different patches denote their resistance to or suitability for the migration of species or the expansion of ecological spaces. Patches with higher *MCR* values are less suitable for ecological expansion and are accordingly defined as having lower ecological suitability levels [61].

### 2.2.2. Model Variables

All the aforementioned analyses were conducted with the help of a cost–distance module in Esri's geographical information system ArcGIS after two essential variables were obtained. The "source" includes the input variables *i* and *j* in Equation (1) and refers to landscape patches that have suitable habitats and provide indispensable ecosystem services; these patches could also be targets in the *MCR* analysis [48,61]. Based on existing research and public documents, the important and ecological protection lands in a major function-oriented zoning (*MFOZ*) region are considered naturally protected regions with significant natural/cultural values and biodiversity [62]. Therefore, we chose the ecological protection lands among the published MFOZ information of Jiangsu as the source; these areas included nature reserves, forest parks, scenic spots, geo-parks, drinking water protection areas (*DWPAs*), flood storage and detention basins (*FSDBs*), fisheries and aquatic resource protection areas (*FARPAs*), and important wetlands, water-dilution channels, ecological forests, and water conservation and reserve areas (*WCRPAs*) (Figure 2a). The cumulative area of these regions is approximately 7055 km$^2$, accounting for 25% of southern Jiangsu.

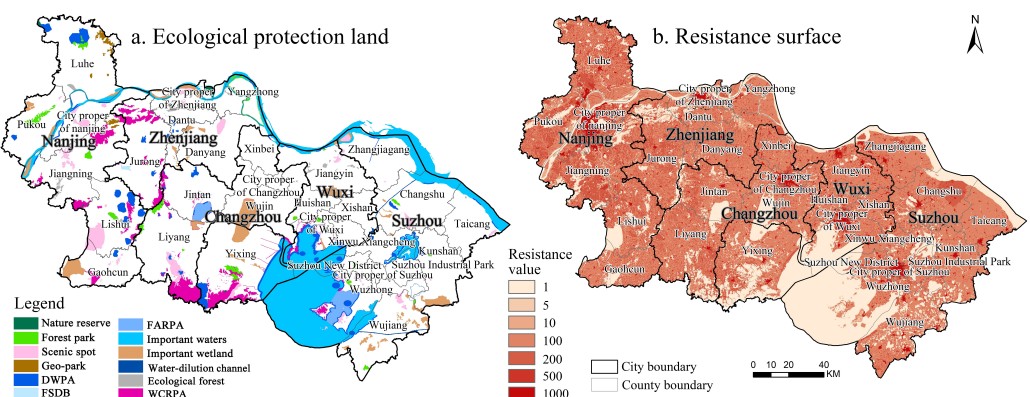

**Figure 2.** Ecological protection lands and resistance surface of southern Jiangsu.

The other necessary variable, i.e., the resistance surface, is designed based on the resistance values of different land use types; these values are relative and have different extents [51,63]. Basically, land use types with higher human disturbance levels, such as greater economies and population agglomeration degrees, usually have greater resistance to the dispersal of common species [48,59]. Li et al. [59] and Liu et al. [53] used the values 1–5 to indicate differences in resistance values, while Su et al. empirically assigned the resistances of different land use types, i.e., 0 for woodlands, 10 for water bodies and shrublands, 30 for gardens and grasslands, 100 for paddy fields, 300 for dried fields, 400 for construction lands and 500 for highways [52]. Adriaensen et al. pointed out that larger relative differences among various land use types are more effective in deviating further from the default straight line between the source and target patches [48]. Based on existing studies and aiming to magnify the different ecological disturbance levels of land use types, we herein assigned resistance values to different land use types, i.e., 1000 for urban lands, 500 for rural settlements and independent industrial and mining land (IIMLs), 200 for dry croplands, 100 for paddy lands and unused lands, 50 for low-coverage grasslands, 30 for moderate-coverage grasslands, 10 for high-coverage grasslands, sparsely forested lands, and other forestlands, 5 for shrublands, and 1 for woodlands and water bodies. Based on land use data collected in 1990, the resistance surface was obtained (Figure 2b).

### 2.2.3. Recognition of Construction Lands with Different Ecological Stress Levels

Based on the *MCR* analysis results, regions with lower *MCR* values are widely accepted to be more suitable for the expansion of ecological spaces and hence, have higher ecological suitability levels [26,53]. To quantitatively evaluate the ecological stress levels of construction lands, it is essential to define the levels of ecological suitability. There were no widely accepted quantitative criteria for determining it. Although more levels could indicate the spatial differentiation of ecological suitability more precisely, it would bring higher complexity accordingly. The recognition of five level suitability was widely accepted, as it is both robust in indicating spatial differentiation and feasible in quantitative calculation [19,53]. Moreover, it was enough to reflect the differences of the ecological stress of construction lands. Therefore, we chose to identify five ecological zones with different ecological suitability levels, i.e., we differentiated zones with highest, moderate-high, moderate, moderate-low, and lowest levels. The zone with the highest *MCR* value had the lowest ecological suitability, as greater disturbances were caused to ecological processes in this region, and vice versa. Reclassification by the natural breaks (Jenks) method was adopted to divide the whole region into subregions with different ecological suitability levels, as this method could minimize differences within groups and maximize differences among groups.

As mentioned above, the ecological stress associated with construction lands is affected by the spatial distribution of the construction lands and the spatial heterogeneity of the ecological suitability level. Basically, construction lands distributed in ecological zones with higher ecological suitability levels cause greater ecological stress. To quantitatively calculate their ecological stress levels, the construction lands in the five designated ecological zones were assigned corresponding stress values, i.e., 5, 4, 3, 2 and 1 for lands in ecological zones with the highest, moderate-high, moderate, moderate-low, and lowest ecological suitability levels.

In this part, maps of both *MCR* values and zones with different ecological suitability levels were obtained. The assigned ecological stress induced by the presence of construction lands at different stages was also identified.

### 2.2.4. Calculation of County-Level Ecological Stress

As administrative districts, especially counties, are the basic units applied for land use regulation and the implementation of related policies, we calculated the general ecological stress levels corresponding to different units. The above analysis allowed the ecological stress associated with construction lands and the amount of construction lands corresponding to each unit in the five established ecological zones (i.e., zones with different ecological stress levels) to be recognized with the help of ArcGIS software. For each unit, as the amounts or proportions of construction lands with different ecological stress levels are quite diverse, their general ecological stress level also differs. The general stress value can be calculated with the following equation:

$$ES_i = \frac{\sum_{j=1}^{m} A_{ij} \times s_j}{\sum_{j=1}^{m} A_{ij}} \tag{2}$$

where $ES_i$ is the general ecological stress of unit $i$ induced by the presence of construction lands; $A_{kj}$ is the area of construction lands associated with ecological stress value $j$; $S_j$ is the ecological stress value of the abovementioned construction lands; and $j$ is the stress level, which varies from 1 to $m$, and $m$ = 5, as five levels are identified in this research. Similarly, the general ecological stress value of southern Jiangsu was also obtained.

As all construction lands were assigned ecological stress values from 1 to 5, the general stress values of the 33 units and the whole of the southern Jiangsu region theoretically ranges from 1 to 5. However, as construction lands are spatially concentrated in reality, the general stress value might also have a specific extent. In this research, the obtained general ecological stress ranged from 1.99 to 4.92, except in the city proper of Suzhou between

2010 and 2015, when there were no incremental construction lands present. We designed a unified classification standard to indicate the spatial differentiation of the county-level ecological stress according to the results calculated for the 33 units at different stages (from 1.88 to 4.92), i.e., <2.00, 2.00–2.75, 2.75–3.50, 3.50–4.25, and 4.25–5.00.

### 2.2.5. County-Level Zoning Based on Both the Area and Ecological Stress Level of Incremental Construction Lands

Both the amount of and ecological stress associated with incremental construction lands are influential for the future land use regulation of a certain unit. Therefore, we carried out county-level zoning after comparing these metrics in a given unit to the regional average. To simplify this process and compare the results, the whole stage from 1990 to 2015 was chosen as the target phase. With 33 units as the input, the average area of incremental construction lands and ecological stress level of southern Jiangsu were 146.04 km$^2$ and 3.81, respectively. Therefore, units with both greater construction land areas and higher stress levels were defined as H-H zones. Similarly, H-L, L-H, and L-L zones were also identified.

### 2.2.6. Data Source and Processing

In this research, we aimed to quantitatively evaluate the ecological stress associated with construction lands and indicate the process and spatial pattern of land use transitions; to this end, land use data representing 1990, 1995, 2000, 2005, 2010 and 2015 were adopted. These data were provided by the Data Center for Resources and Environmental Sciences (RESDC) at the Chinese Academy of Sciences (http://www.resdc.cn, accessed on 22 May 2021) and were interpreted based on Landsat-Thematic Mapper (TM) images at high precision. With land use data from 1990 to 2015, the expansion and spatial distribution of incremental construction lands were obtained in five stages, i.e., 1990–1995, 1995–2000, 2000–2005, 2005–2010 and 2010–2015. Additionally, administrative division data were also provided by RESDC. The sources input into the *MCR* model, i.e., the ecological protection lands, were obtained from published MFOZ data representing Jiangsu.

## 3. Results

### 3.1. Spatial Distribution of Zones with Different Ecological Suitability Levels

The spatial distribution of *MCR* values was similar to the irregular spread of contour lines. Basically, regions close to cities had higher MRCs, as more construction lands were distributed in these regions, while water bodies and mountains/hills had lower *MCR* values (Figure 3a). After reclassification, the five ecological zones with different *MCR* values and ecological suitability levels accounted for 49.85%, 25.73%, 15.56%, 6.51%, and 2.34% of the zones with the highest to lowest ecological suitability levels, respectively. The most suitable zones (with the highest ecological suitability levels and lowest *MCR* values) were mainly concentrated in areas surrounding Taihu Lake and the Yangtze River as well as in the Yixing-Liyang and Mao mountainous regions, which are closer to ecological lands and have low anthropogenic disturbance levels (Figure 3b). The ecological zones with the lowest suitability levels were concentrated in urbanized areas, including in the cities of Nanjing, Suzhou, Wuxi, and Changzhou and in eastern regions close to the Shanghai metropolitan area (Figure 3b). These regions are characterized by widely distributed construction lands, dense populations and industrial agglomerations.

### 3.2. Spatial Distribution of Incremental Construction Lands

After 1990, construction lands in southern Jiangsu expanded drastically, but showed an obvious difference among different growth stages. The expansion of construction lands was 974.65, 456.49, 767.18, 2123.28 and 496.55 at stages from 1990 to 1995, from 1995 to 2000, from 2000 to 2005, from 2005 to 2010, and from 2010 to 2015, respectively. The period from 2005 to 2010 had the fastest growth, when the boost of export-oriented economy caused rapid industrial development and the inflow of floating population as China entered WTO. Hereafter, the expansion decreased obviously as some restrictive

policies on land development were proposed under the background of high land use intensity. Basically, the expansion of construction lands at early stages was basically promoted by the socioeconomic development, while the slowdown in recent years was led by the restriction of land development and the protection of ecological lands and crop lands.

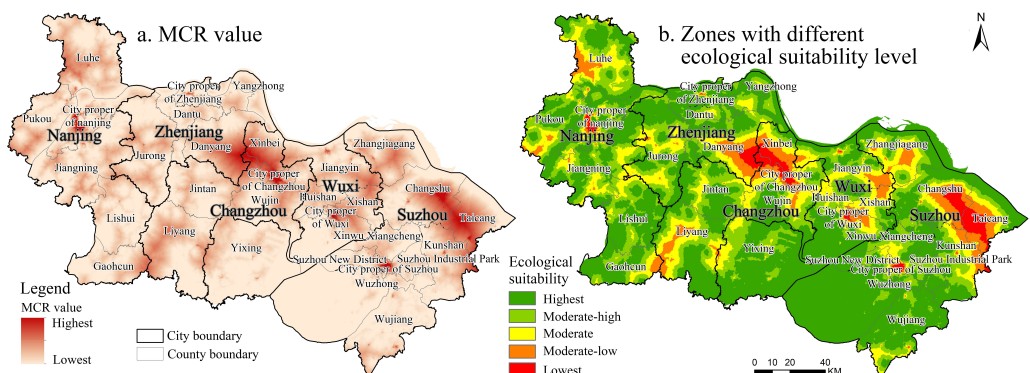

**Figure 3.** *MCR* values and zones with different ecological suitability levels in southern Jiangsu.

Among the five prefectural cites, Suzhou experienced the fastest construction land growth, reaching 1892 km². It accounted for 39.29% of the total increment in southern Jiangsu. This is basically consistent with its proportion of economy and population, i.e., 40.58% and 35.22%, respectively. Suzhou is closest to Shanghai in southern Jiangsu and is also among the most developed cities in China, causing greater demand for construction lands. In terms of county-level units, the cities and surrounding units of Nanjing, Suzhou, Wuxi and Changzhou, as well as Jiangyin, Zhangjiagang, Taicang and Kunshan along the Yangtze River, had greater expansion than other units (Figure 4). The expansion of Kunshan was the largest, at approximately 13 times that of the city proper of Wuxi, the unit with the lowest expansion. The expansion of the cities of Jiangning in Nanjing, Changshu, Wujiang and Zhangjiagang in Suzhou, Jiangyin and Yixing in Wuxi and Wujin in Changzhou were also great, with expansion amounts over 200 km². In contrast, the expansion of Gaochun in Nanjing and the cities of Suzhou and Wuxi were smaller, with expansion amounts less than 50 km². In general, the units around Taihu Lake and along the Yangtze River dominated the expansion of construction lands in southern Jiangsu after 1990. This is also consistent with the pattern of population agglomeration and industrial development.

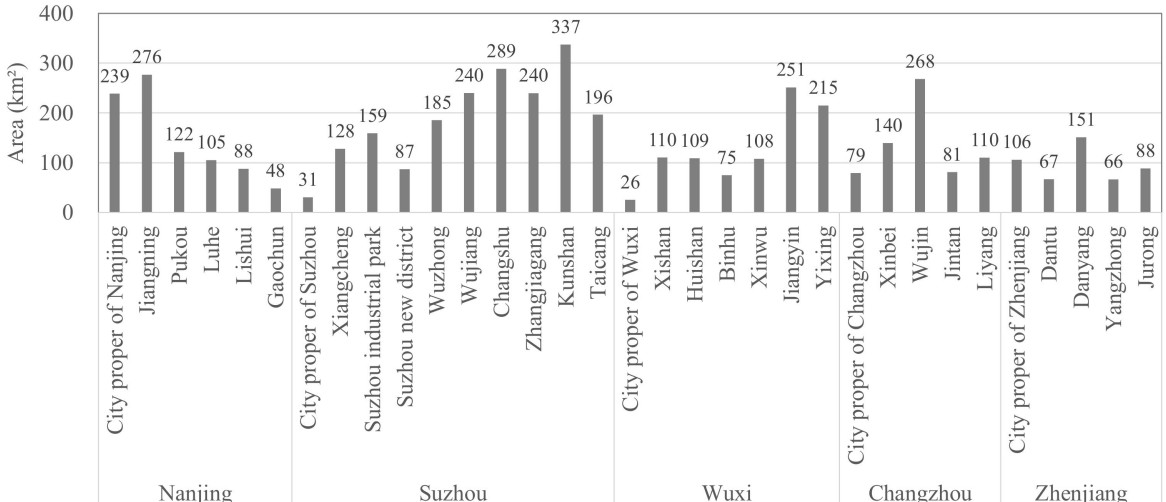

**Figure 4.** The expansion of construction lands in different units from 1990 to 2015.

*3.3. Spatial Differentiation of Incremental Construction Lands with Different Ecological Stress Levels*

In total, our overlay analysis of ecological zones and construction lands indicated that the construction lands with the lowest, moderate-low, moderate, moderate-high, and highest (from level 1 to 5) ecological stress values (i.e., those distributed in ecological zones with ecological suitability levels of 1 to 5) were 153.52, 433.51, 1007.22, 1792.62, and 1431.29 $km^2$, respectively. The majority of these areas had highest and moderate-high ecological stress levels, while areas with the lowest ecological stress made up the smallest fraction.

The ecological stress associated with incremental construction lands showed similar patterns in each stage (Figure 5). There were more lands with highest or moderate-high stress levels than the other categories, while lands with the lowest stress levels made up the smallest fraction. However, there were still slight differences among different stages. From 1990 to 1995, the area of construction lands with moderate-high stress levels was greater than the areas of construction lands with highest and moderate stress levels, but construction lands with the highest stress levels were slightly larger than those with moderate-high stress levels from 1995 to 2000. From 2000 to 2005 and from 2005 to 2010, the area of lands with moderate-high stress levels again became dominant. From 2010 to 2015, the largest fraction was again replaced by lands with the highest stress level. In general, construction lands with highest and high stress levels alternately dominated the incremental construction lands, indicating that the ecological stress caused by incremental construction lands remained at a high level in each stage.

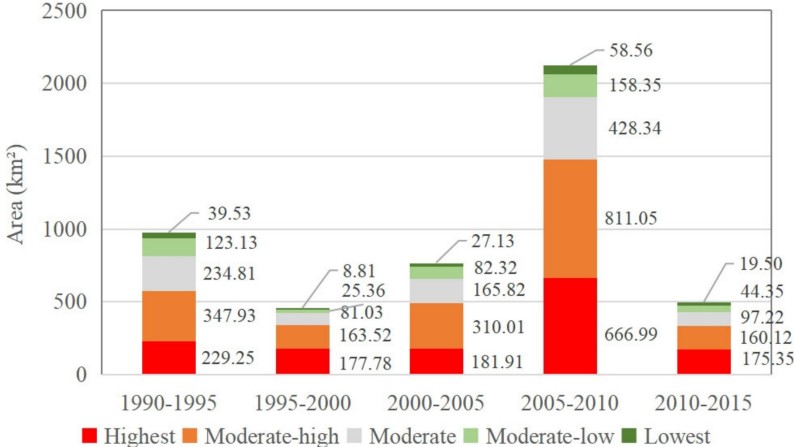

**Figure 5.** Areas of incremental construction lands associated with different ecological stress levels.

Spatially, the construction lands with the highest ecological stress levels were mainly distributed in areas along the Yangtze River and around Taihu Lake (Figure 6a). Wujiang, Wuzhong and Binhu around Taihu Lake and Jiangyin, Zhangjiagang, and the cities of Zhenjiang and Nanjing along the Yangtze River had more lands with the highest stress levels. Moreover, some county-level units with high proportions of waters, hills or mountains also had large land areas with the highest stress levels, including Suzhou Industrial Park, Lishui, Kunshan, Pukou, Jiangning and Yixing. In contrast, the lowest-stress construction lands were mainly distributed in the surrounding units of cities, such as around the city proper of Changzhou and the adjacent Xinbei, the city proper of Wuxi and the adjacent Xishan, Xinwu, and Taicang, and Changshu and Kunshan in the eastern part of southern Jiangsu. There were fewer ecological lands and with lower ecological suitability, and the lower ecological stress was caused by construction lands.

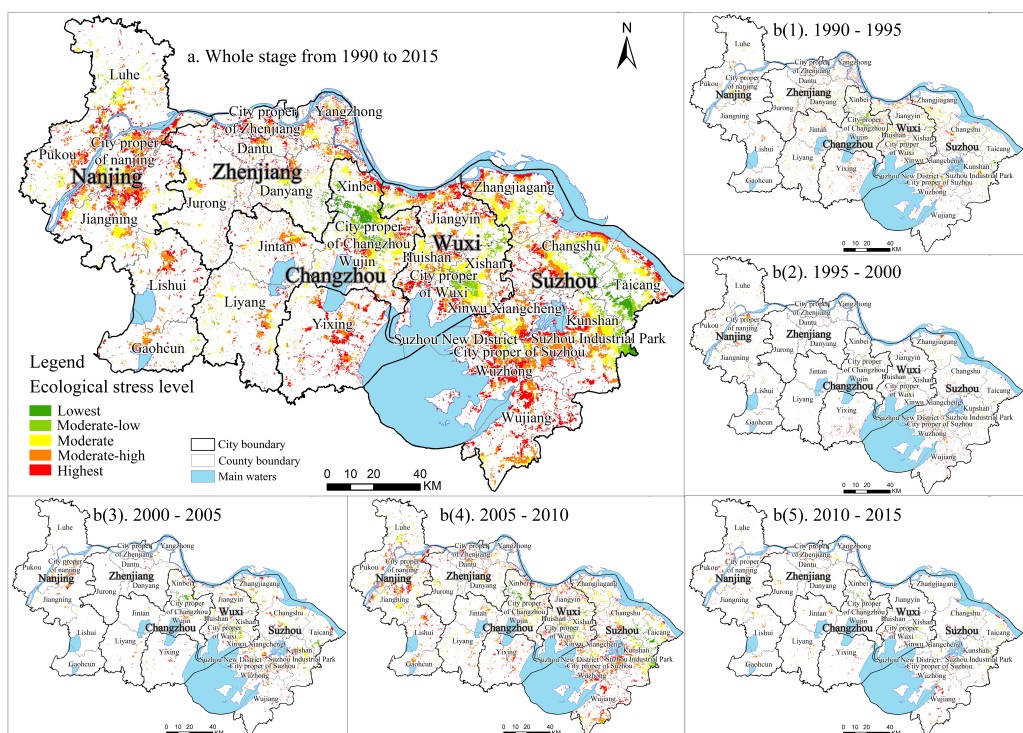

**Figure 6.** Spatial distribution of construction lands associated with different ecological stress levels in each stage.

The spatial differentiation of construction lands associated with different ecological stress levels showed similar patterns in each stage (Figure 6b). Areas associated with high ecological stress levels were mainly concentrated at the units along the Yangtze River, around Taihu Lake and in mountainous areas, while construction lands associated with relatively low stress levels were basically distributed at cities or in some counties with low ecological suitability levels, including in southern Taicang, eastern Changshu and northwestern Kunshan.

### 3.4. Patterns of Units with Different Ecological Stress Levels

From 1990 to 2015, the general ecological stress level associated with incremental construction lands was 3.81, characterizing a relatively high level. This is because the majority of construction lands were of highest and moderately high ecological stress levels. As this value is affected by the spatial differentiation of incremental construction lands associated with different ecological stress levels, the average ecological stress level associated with each unit differed significantly (Figure 7a). Among the units, the ecological stress of the city proper of Changzhou was low (i.e., 2.38), while that of Binhu of Wuxi was highest (i.e., 4.67) and was approximately 1.96 times that of the city proper of Changzhou. Based on the reclassification results, eight units had ecological stress levels higher than 4.25, and these units were mainly distributed around Taihu Lake and along the Yangtze River, including Binhu and Yixing of Wuxi, Wuzhong and Wujiang of Suzhou, Dantu, Yangzhong and the city proper of Zhenjiang, and Lishui of Nanjing. Conversely, there were no units with stress levels lower than 2.00. The stress levels of 3 units were between 2.00 and 2.75, and these units were mainly distributed along the Yangtze River, including Taicang of Suzhou, Xinbei, and the city proper of Changzhou. Units with stress levels between 3.50 and 4.25 dominated the whole region, and the total number of units falling within this range was 17, further indicating that the ecological stress associated with incremental construction lands was at a relatively high level.

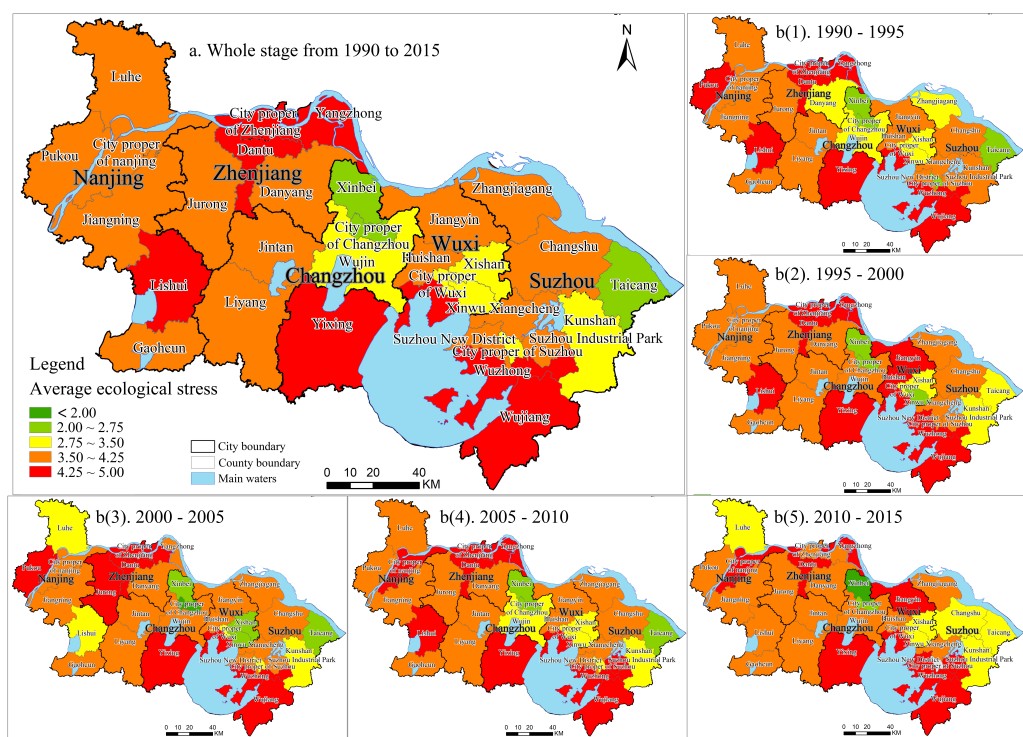

**Figure 7.** Spatial distributions of units with different ecological stress levels in each stage.

Since 1990, the general ecological stress levels determined at each stage were 3.62, 4.04, 3.70, 3.88 and 3.86, showing a fluctuation trend of "increasing-decreasing-increasing". The spatial pattern of each stage showed similar characteristics to that of the whole stage, i.e., units with moderate-high stress levels dominated southern Jiangsu, followed by those with the highest and moderate-low stress levels (Table 1; Figure 7b). Moreover, few units had ecological stress levels below 2.00, including the city proper of Changzhou from 2000 to 2005 and Xinbei in Changzhou from 2010 to 2015. Specifically, as the construction lands in the city proper of Suzhou remained stable from 2010 to 2015, the ecological stress level of this unit was 0.

**Table 1.** Numbers of units with different ecological stress levels in each stage.

| Stage | Lowest <2.00 | Moderate-Low 2.00~2.75 | Moderate 2.75~3.50 | Moderate-High 3.50~4.25 | Highest 4.25~5.00 |
|---|---|---|---|---|---|
| 1990–1995 | 0 | 4 | 6 | 14 | 9 |
| 1995–2000 | 0 | 3 | 4 | 15 | 11 |
| 2000–2005 | 1 | 4 | 5 | 15 | 8 |
| 2005–2010 | 0 | 3 | 6 | 14 | 10 |
| 2010–2015 | 2 | 1 | 6 | 14 | 10 |
| Total | 0 | 3 | 6 | 16 | 8 |

*3.5. Zones with Different Amounts of and Ecological Stress Levels Associated with Incremental Construction Lands*

As determined from the relationship between the amount of incremental construction lands and the average stress level associated with these lands in each unit, Moran's I value obtained from the local indicators of spatial association (LISA) analysis was −0.1531, indicating that units with greater expansion areas had lower average stress values at the county scale. The more construction lands expanded from 1990 to 2015 in a unit, the lower the ecological stress level of that unit. According to the relationship between the expansion amount of each unit and its average ecological stress level, four types of zones were identified further (Figure 8).

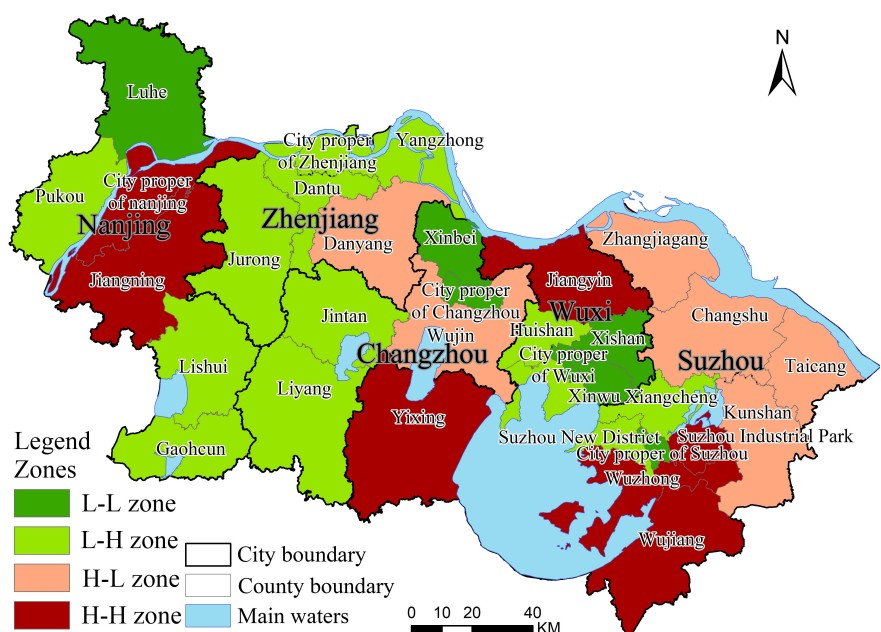

**Figure 8.** Zones with different construction land expansion extents and their corresponding ecological stress levels.

(1) H-H zones. These zones included units with both greater construction land expansion amounts and higher ecological stress levels than the corresponding regional averages, comprising Jiangning of Nanjing, Jiangyin and Yixing of Wuxi, and Suzhou Industrial Park, Wuzhong and Wujiang of Suzhou. These units were rapidly developing areas with significant populations and industry agglomerations after 1990, causing the remarkable expansion of construction lands. As they were also zones with higher ecological suitability, the ecological stress caused by construction lands was at a high level. Taking the city proper of Nanjing as an example, from 1990 to 2015, its population increased from 2.32 million to 5.26 million, and its GDP increased from RMB 12.34 billion to 444.58 billion. However, as water bodies and mountains are also widely distributed in this unit, they contributed a higher ecological suitability level. Therefore, the rapid expansion of construction lands brought more serious disturbances to ecological safety, and a higher ecological stress level was caused accordingly.

(2) H-L zones. These zones comprised units with both less construction land expansion and lower ecological stress levels than the corresponding regional averages, including Danyang of Zhenjiang, Wujin of Changzhou, and Kunshan, Taicang, Changshu and Zhangjiagang of Suzhou. These units contain developed areas with rapidly growing populations and industry agglomeration after 1990, causing significant construction land expansion. Kunshan was a typical case; in this city, the population increased from 0.56 million to 1.65 million and the GDP increased from RMB 1.86 billion to 308.00 billion. As the above regions were also less ecologically suitable, the growth of a large amount of construction lands did not cause high ecological stress levels.

(3) L-H zones. These zones included units with less of construction land expansion but higher ecological stress levels than the regional averages. There were 13 units in this category in total, accounting for 40% of all units, mainly including units surrounding cities or in ecologically suitable regions. Although the expansion construction lands were relatively small in these units, the ecological stress levels were higher than the regional average. The construction lands of the city proper of Zhenjiang increased by approximately 106 km$^2$, as its proportion of construction lands was high before 1990 and expansion of construction lands was restricted accordingly. However, as this city is located along the Yangtze River with a high ecological suitability level, its ecological stress level was 4.41, much higher than the regional average (i.e., 3.81).

(4) L-L zones. This category included units with both less construction land expansion and lower ecological stress levels than the corresponding regional averages and comprised Luhe of Nanjing, the city proper of Xinbei of Changzhou, the city proper of Suzhou, and the cities of Xinwu and Xishan of Wuxi. This category mainly included cities or their adjacent units. As mentioned above, these areas already had very high proportions of construction lands before 1990, as they were urban centers with significant populations and industry agglomeration in the early stage. With the increasing area of incremental construction lands located in suburban areas, land development was limited. Meanwhile, early land development reduced the ecological suitability level, so the subsequent expansion of construction lands did not contribute to extensive ecological stress. Taking the city proper of Suzhou as an example, the construction lands in this city expanded by merely 31 km$^2$, while the ecological stress level was 2.96, almost the lowest value determined among all units.

## 4. Discussion

Rapid, intensive urbanization and the spatial expansion of construction lands introduce significant ecological stress to regional eco-safety and sustainable development in various ways [64,65]. It is among the topics of focus for academics to indicate the processes and patterns of land use transitions by evaluating the ecological stress levels associated with land use changes, especially with the expansion of construction lands [19]. Although different attributes of construction lands influence the resulting ecological stress levels, including the amount, proportion, detailed type and concentration degrees of industries and populations, the location effect is receiving the most attention [66,67]. Aiming to determine the pattern of land use transitions based on the ecological stress levels related to the spatial distribution of construction lands, we proposed an improved approach to evaluate the ecological stress level from the perspective of the spatial distribution of incremental construction lands. Taking the economically developed and densely populated southern Jiangsu as the case, we carried out ecological process simulations to determine the spatial differentiation of ecological suitability, identified the spatially variable ecological stress levels associated with incremental construction lands based on their distributions, and recognized different zones by comparing the incremental construction land areas and the associated ecological stress levels.

To indicate the spatial differentiation of the ecological stress contributed by construction lands, it is essential to analyze the ecological suitability levels of different regions. In regions with higher ecological suitability levels, the presence of construction lands generally induces higher ecological stress levels [26,68]. Although various methods have been widely adopted to evaluate ecological suitability, we chose to study suitability from the perspective of maintaining the integrity of ecological processes, as unbroken ecological processes are of the utmost importance for the conservation of biodiversity and ecosystem services [27,33]. With the support of the GIS spatial analysis function, we constructed ecological process simulations with the help of the *MCR* model due to its superiority in combining GIS spatial analyses with map making and operability in the data preparation process [48]. Based on the identification of the sources and resistance surface, the potential ecological processes and zones with different ecological suitability levels in southern Jiangsu were revealed. The results indicated that the spatial differentiation of the indicated ecological suitability was basically consistent with the physical geography and ecosystem features. The spaces surrounding certain sources/targets were of higher ecological suitability levels, while the built-up areas of cities were less suitable. Suitable ecological zones appeared not only as patches, but also as corridors along suitable landscape patches, especially along some continuous rivers. The importance of these indicated corridors in maintaining regional ecological safety was revealed, as these corridors could increase the connectivity of patched ecological spaces [69,70].

The spatial expansion of construction lands was dramatic and showed significant spatiotemporal differences from the perspective of their amount and spatial locations. This result was consistent with the economic development and population inflows in terms of the analyzed stages, and the period from 2005 to 2010, when the construction lands expanded the most, was also the period with the fastest economic growth and greatest population agglomeration. Spatially, the relationship between construction land expansion and socioeconomic development was also highly positive. From the perspective of the associated ecological stress, construction lands in zones with higher ecological suitability levels brought more significant disturbances to the ecological processes and hence, contributed to higher ecological stress levels. Basically, the incremental construction lands corresponded to high ecological stress levels, and the average stress value was 3.81 with a maximum of 5.00, indicating that the spatial expansion of construction lands was not ecologically friendly on the whole. From a quantitative perspective, only a limited area of incremental construction lands was associated with low ecological stress levels. It is suggested that the spatial expansion of construction lands should be restricted in southern Jiangsu [71,72]. However, as the spatial location of incremental construction lands affected the ecological stress value, these regulations should be spatially variable. In this study, incremental construction lands associated with high ecological stress levels were widely distributed and were mostly concentrated in some regions along Taihu Lake and the Yangtze River. As strong interference with ecological processes occurs, construction concessions and ecological restoration should be conducted in these regions [73]. However, such regions were also the preferred areas for land development and industrial growth [36,74]. To balance the relationship between ecological protection and land development, construction lands should be expanded while trying to control the occupation of areas with the highest ecological suitability levels, especially those that form ecological corridors [73,75]. Existing construction lands with high ecological stress levels should be ecologically restored as they occupy strategic points or potential corridors of regional ecological processes [25,27].

As counties are the basic units of governmental regulation regarding land development in China, we further derived ecological stress results at the county level. Although the spatial differentiation of county-level units with different ecological stress levels was slightly variable among each stage, units with moderate-high and highest stress levels dominated the study. As confirmed by the above results, a large amount of incremental construction lands is associated with high ecological stress levels. Basically, units with high ecological stress levels were concentrated around Taihu Lake and along the Yangtze River, where the ecological suitability was higher than that in other regions [19]. As both the expansion amount and spatial distribution of construction lands were related to different ecological stress levels, we identified four types of zones by comparing their expansion areas and ecological stress levels with the corresponding regional average values obtained for the whole study area. The H-H zones had both greater construction land areas and higher stress values; thus, it is suggested that the expansion of construction lands in these zones be strictly controlled to decrease ecological disturbances. Moreover, this is also the focus of ecological restoration, as the presence of a large amount of incremental construction lands induces greater ecological stress. For the H-L zones, although the land development was also fast, the resulting ecological stress was relatively low. Therefore, further land development should be encouraged through supportive land policies. The L-H zones caused high ecological stress with limited incremental construction lands, so further land development in these zones should be moved to regions with low ecological suitability levels [68,76]. Although the ecological stress level of the L-L zones was also low, land development should also be restricted in these regions because they already have high proportions of construction lands. Further land development in these regions is highly likely to occupy ecologically suitable spaces and destroy regional ecological safety.

### 5. Conclusions

The ecological protection and expansion of construction land are both location related, and there are spatial conflicts between them. The ecological stress caused by construction lands differs spatially, and these differences are caused by both the spatial locations of construction lands and the location-related differentiation of ecological suitability. Aiming to promote sustainable development and coordinate the relationship between land development and ecological protection, with the help of the *MCR* model and ecological process simulations, we identified the spatial differentiation of ecological suitability levels. Then, we evaluated the ecological stress levels of incremental construction lands and county-level zoning regions based on the amount of and ecological stress level associated with local incremental construction lands during different stages of land use transitions. We conclude that the location of incremental construction lands significantly affects the ecological consequences associated with the spatial differentiation of ecological suitability. Basically, incremental construction lands located in ecologically suitable zones cause high ecological stress. As the indicated ecological suitability level is obtained through an ecological process analysis, the location-specific ecological stress level has an obvious influence on regional ecological safety from the perspective of maintaining ecosystem integrity. Both construction lands and ecological suitability are location specific, and we believe that location-oriented evaluations could provide an effective approach for determining the spatial patterns of land use transitions based on spatially differentiated ecological consequences. According to these results, location-specific suggestions regarding land use regulations could also be proposed, including suggestions for spatially precise ecological restoration and the spatial redistribution of incremental construction lands.

**Author Contributions:** Conceptualization, P.L.; data curation and analysis, C.L.; writing—original draft preparation, P.L. and H.C. All authors have read and agreed to the published version of the manuscript.

**Funding:** This study was funded by the National Natural Science Foundation of China (Grant Nos. 41871209 and 41901215) and the Strategic Priority Research Program of Chinese Academy of Sciences (grant No. XDA23020102).

**Institutional Review Board Statement:** Not applicable.

**Informed Consent Statement:** Not applicable.

**Data Availability Statement:** The data presented in this study are available on request from the corresponding author.

**Conflicts of Interest:** The authors declare no conflict of interest.

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
