# Peer review of "Quantitative Evaluation of Ecological Stress Caused by Land Use Transitions Considering the Location of Incremental Construction Lands: The Case of Southern Jiangsu in Yangtze River Delta Region"

_land, doi:10.3390/land11020175_

Round 1

Reviewer 1 Report

This study used the MCR model and ecological process simulation, carried out the quantitative evaluation of ecological pressure of land use transformation in southern Jiangsu, and discussed the relationship between incremental construction land and ecological suitability level, which has certain practical significance. However, the paper requires substantial changes before it can be accepted. In particular, the logic of the article needs to be sorted out to make it more readable. I have the following suggestions for the authors to improve the quality of the study:

1、Land urbanization mentioned in the title is rarely mentioned in the paper. Whether the use of keywords in urbanized region is appropriate still needs to be considered.

2、In the abstract part, I suggest adjusting to the paradigm of "research significance+ research ideas + research results + conclusions". In your summary, I can't get the results of the research.

3、Line 36: I suggested adding a reference to explain the source of the data involved in this sentence.

4、Line 41-45: On the expression of research motivation and purpose, because the current research situation shows that it is important, we should study it? I suggest sorting out the research motivation.

5、Gaps and originality are unclear. Does the novelty consist in the location factors? Please highlight the study's contribution.

6、In the study area section, I suggested using dynamic change data instead of current situation data. I can't understand the range of land use transition in the study area from your expression. The reasons for choosing southern Jiangsu as the study area rather than other areas should be explained clearly.

7、Section 3.2 and other results showing section should interpret the reasons for the expansion of construction land from 2005 to 2010. Another example is the reason for the fastest growth of construction land in Suzhou.

8、The basis for the establishment of five ecological areas with different MCR values and ecological suitability levels should be explained. In other words, although the natural breakpoint method is adopted, the basic characteristics of five types of zoning should be clarified.

9、The manuscript has always emphasized the location effect, but I feel that the location effect has not been fully reflected after reading the full text.

10、The format of references needs to be adjusted.

11、Some small details : (1) Line 104 and Line 441: The word “transition” or “transitions”? (2) Line 323-324: In typesetting, pay attention to the drawing and title, and do not spread across pages. (3) Line 286:Figure 4 can be adjusted for viewing. I suggest deleting redundant elements.

Author Response

Revision Report for Manuscript the ID: LAND-1530883

Quantitative Evaluation of Ecological Stress Caused by Land Use Transitions Considering the Location of Incremental Construction lands: The Case of Southern Jiangsu in Yangtze River Delta region

January, 7, 2022

Dear Reviewer,

We would like to thank you for giving us the opportunity to revise our manuscript. We have taken this opportunity very seriously and made great efforts to revise our manuscript based on your comments. All crucial revising contents are marked in red in the manuscript file.

Sincerely yours,

The author team

Responses

Comment 1:

This study used the MCR model and ecological process simulation, carried out the quantitative evaluation of ecological pressure of land use transformation in southern Jiangsu, and discussed the relationship between incremental construction land and ecological suitability level, which has certain practical significance. However, the paper requires substantial changes before it can be accepted. In particular, the logic of the article needs to be sorted out to make it more readable. I have the following suggestions for the authors to improve the quality of the study.

Response:

Thank you for your approval and suggestions. We made detailed modifications to our manuscript to better sort out the logic and make it more readable.

Comment 2:

Land urbanization mentioned in the title is rarely mentioned in the paper. Whether the use of keywords in urbanized region is appropriate still needs to be considered.

Response:

Thank you for your suggestion. Land urbanization refers to the evolution of land-use attributes and surface landscape from rural to urban land. In the manuscript, we usually used the expansion of construction lands to represent it. To avoid the confusion of the vocabularies, we changed the “Land urbanization” to “Incremental Construction Lands”. We also changed the “urbanized region” to “southern Jiangsu, China” as one of the key words to indicate the case area of this research. Moreover, “location” was also added as a key word to emphasize the research topic and motivation.

Comment 3:

In the abstract part, I suggest adjusting to the paradigm of "research significance+ research ideas + research results + conclusions". In your summary, I can't get the results of the research.

Response:

Thank you for your suggestion. We re-wrote the abstract according to the paradigm of "research significance + research ideas + research results + conclusions" and also added more contents about the results. Moreover, the research topic and innovation was also emphasized, and “location” and “southern Jiangsu, China” were added as key words. See P1, L13-40.

Comment 4:

Line 36: I suggested adding a reference to explain the source of the data involved in this sentence.

Response:

Thank you for your suggestion. Considering the data was provided in China Statistical Year Books, we added them as the data sources (P2, L46).

Comment 5:

Line 41-45: On the expression of research motivation and purpose, because the current research situation shows that it is important, we should study it? I suggest sorting out the research motivation.

Response:

Thank you for your suggestion. We rephrased the ABSTRACT and made modifications to the INTRODUCTION to emphasize the research topic and motivation. The spatial conflict between ecological protection and expansion of construction lands existed and brought challenges to regional sustainable development. However, the spatial differentiation of ecological suitability and related ecological stress of construction lands during different stages of land use transitions have not been analysed. To achieve it, we first identified the spatial heterogeneity of ecological suitability from the perspective of ecological process analysis, and then quantitatively assessed the ecological stress of construction lands at different stages after 1990. Based on the results of ecological stress evaluation of construction lands and following zoning, we proposed location-specific policies to carry out spatially precise ecological restoration and the redistribution of incremental construction lands. Related contents were added in the ABTSRACT (P1, L13-18) and INTRODUCTION (P2, L51-55, L66-78; P2-3, L81-99; P3, L127-144).

Comment 6:

Gaps and originality are unclear. Does the novelty consist in the location factors? Please highlight the study's contribution.

Response:

Thank you for your suggestion. The analysis of location effect was the topic of this research. As mentioned in the response to Comment 5, the main topic is to indicate the spatial conflict between ecological protection and expansion of construction land, which are both location-related. To achieve it, we first identified the spatial heterogeneity of ecological suitability from the perspective of ecological process analysis, and then quantitatively assessed the ecological stress of construction lands at different stages after 1990. Based on the results of ecological stress evaluation of construction lands and following zoning, we proposed location-specific policies to carry out spatially precise ecological restoration and the redistribution of incremental construction lands. During the modification, we rephrased the whole ABSTRACT and part of the INTRODUCTION to emphasize the research topic and contribution (P1, L13-18; P2-3, L81-99; P3, L175-144). Related contents also appeared in the CONCLUSION (P15, L605-615). Moreover, Location was added as one of the key words, and we greatly increased its appearance number of times to 34 in the revised manuscript.

Comment 7:

In the study area section, I suggested using dynamic change data instead of current situation data. I can't understand the range of land use transition in the study area from your expression. The reasons for choosing southern Jiangsu as the study area rather than other areas should be explained clearly.

Response:

Thank you for your suggestion. We added the growth of regional residents and GDP and the expansion of construction lands in southern Jiangsu from 1990 to 2015. Moreover, related literatures were also cited to better reflect its socio-economic development and land use transitions. We hope we can better explain why southern Jiangsu was chosen as the study area. See P4, L154-175.

Comment 8:

Section 3.2 and other results showing section should interpret the reasons for the expansion of construction land from 2005 to 2010. Another example is the reason for the fastest growth of construction land in Suzhou.

Response:

Thank you for your suggestion. We added necessary reasons in the required sections. See P8, L344-355, L371-372; P10, L408-410, L423-424; P12, L465-469, L491-493, L500-504; etc.

Comment 9:

The basis for the establishment of five ecological areas with different MCR values and ecological suitability levels should be explained. In other words, although the natural breakpoint method is adopted, the basic characteristics of five types of zoning should be clarified.

Response:

Thank you for your suggestion. We added necessary descriptions in the revised manuscript (See P6, L247-258). We pointed out: To quantitatively evaluate the ecological stress levels of construction lands, it is essential to define the levels of ecological suitability. There were no widely accepted quantitative criteria for determining it. Although more levels could indicate the spatial differentiation of ecological suitability more precisely, it would bring higher complexity accordingly. The recognition of five level suitability were widely accepted as it is both robust in indicating spatial differentiation and feasible in quantitative calculation. Moreover, it was enough to reflect the differences of the ecological stress of construction lands. Therefore, we chose to identify five ecological zones with different ecological suitability levels, i.e., we differentiated zones with highest, moderate-high, moderate, moderate-low, and lowest levels. The zone with the highest MCR value had the lowest ecological suitability, as greater disturbances were caused to ecological processes in this region, and vice versa.

Comment 10:

The manuscript has always emphasized the location effect, but I feel that the location effect has not been fully reflected after reading the full text.

Response:

Thank you for your suggestion. As mentioned by this comment, location effect was emphasized as the main topic of our research. It is our deficiency for not making it significant enough. In the revised manuscript, we greatly increased its appearance number of times to 34. The word “Location” was added in the TITLE and also listed as one of the key words. Moreover, we greatly changed the ABSTRACT and INTRODUCTION together with the responses to the Comment 5 and Comment 6, and emphasized the research on location effect during sorting out the research motivation and novelty. See P2, L66-78; P2-3, L81-99; P3, L127-144.

Comment 11:

The format of references needs to be adjusted.

Response:

Thank you for your suggestion. We adjusted the format of all references in detail according to the requirement of LAND.

Comment 12:

Some small details: (1) Line 104 and Line 441: The word “transition” or “transitions”? (2) Line 323-324: In typesetting, pay attention to the drawing and title, and do not spread across pages. (3) Line 286:Figure 4 can be adjusted for viewing. I suggest deleting redundant elements.

Response:

Thank you for your detailed suggestions. We made following modifications:

(1) We changed “transition” to “transitions” to better fit the contents and also made necessary changes to related sentences.

(2) We changed the format of Figure 4 and Figure 5 (i.e. Figure 5 and Figure 6 in the original manuscript) to make sure the drawing and title do not spread across pages (P9, L373 and L396).

(3) We deleted Figure 4 as what this picture is trying to express can be described in one sentence (P8, L344-348). The number of the following figures were changed accordingly.

Reviewer 2 Report

This is a standard well written paper. Considering the high-quality standard of land, I would suggest some revisions before acceptance. For me, the main issue in the paper is that it is routinary and very normal level. This is not bad per sè, but I think readers would read some motivations about the originality of the paper. So I strongly encourage authors to enrich the comments about novelty of their approach, representativeness of the study area, generalization of the results to other contexts. Only giving value to the article, motivating novelty and exceptionality of this study, will be possible to make a thorough contribution to science. In this perspective, language usage can be checked in some parts of the text (mainly introduction). AUthors are free to enrich also their literature review, if necessary.

Author Response

Revision Report for Manuscript the ID: LAND-1530883

Quantitative Evaluation of Ecological Stress Caused by Land Use Transitions Considering the Location of Incremental Construction lands: The Case of Southern Jiangsu in Yangtze River Delta region

January, 7, 2022

Dear Reviewer,

We would like to thank you for giving us the opportunity to revise our manuscript. We have taken this opportunity very seriously and made great efforts to revise our manuscript based on your comments. All crucial revising contents are marked in red in the manuscript file.

Sincerely yours,

The author team

Responses

Comment 1:

This is a standard well written paper. Considering the high-quality standard of land, I would suggest some revisions before acceptance.

Response:

We thank you for your approval!

Comment 2:

For me, the main issue in the paper is that it is routinary and very normal level. This is not bad per sè, but I think readers would read some motivations about the originality of the paper. So I strongly encourage authors to enrich the comments about novelty of their approach, representativeness of the study area, generalization of the results to other contexts. Only giving value to the article, motivating novelty and exceptionality of this study, will be possible to make a thorough contribution to science. In this perspective, language usage can be checked in some parts of the text (mainly introduction). Authors are free to enrich also their literature review, if necessary.

Response:

Thank you for your suggestion. We completely agree with this comment. We rephrased the ABSTRACT and made modifications to the INTRODUCTION to emphasize the research topic and innovation. The main topic is to indicate the spatial conflict between ecological protection and expansion of construction land. To achieve it, we first identified the spatial heterogeneity of ecological suitability from the perspective of ecological process analysis, and then quantitatively assessed the ecological stress of incremental construction lands at different stages after 1990. Based on the results of ecological stress evaluation and following zoning, we proposed location-specific policies to carry out spatially precise ecological restoration and the redistribution of incremental construction lands. Related contents were added in the ABTSRACT (P1, L13-18) and INTRODUCTION (P2, L51-55, L66-78; P2-3, L81-99; P3, L127-144), and also the DISCUSSION (P14, L550-551, L556-559, L565-578) and CONCLUSION (P15, L605-615).

Together with the modification, more than ten related papers were added and reviewed to support above viewpoint (See references marked in red).

Round 2

Reviewer 2 Report

Accurate revision overall, thank you.